# Compassion Satisfaction, Burnout, and Secondary Traumatic Stress among Respiratory Therapists in Mississippi: A Cross-Sectional Study

Driscoll DeVaul [1], Britney Reulet [1], Jacob Daniels [2], Xiaoqian Zhu [3], Renee Wilkins [4] and Xiaoshan Z. Gordy [1,*]

[1] Department of Health Sciences, School of Health Related Professions, University of Mississippi Medical Center, Jackson, MS 39216, USA; ddevaul@umc.edu (D.D.); breulet@umc.edu (B.R.)
[2] Department of Physical Therapy, School of Health Related Professions, University of Mississippi Medical Center, Jackson, MS 39216, USA; jbdaniels@umc.edu
[3] Department of Data Science, School of Population Health, University of Mississippi Medical Center, Jackson, MS 39216, USA; xzhu3@umc.edu
[4] Department of Medical Laboratory Science, School of Health-Related Professions, University of Mississippi Medical Center, Jackson, MS 39216, USA; rwilkins@umc.edu
* Correspondence: xgordy@umc.edu

**Abstract:** The COVID-19 pandemic had an immense effect on the well-being of healthcare professionals. In this study, researchers utilized a quantitative cross-sectional study design to investigate the degree of compassion satisfaction and fatigue amongst respiratory therapists in the state of Mississippi as a result of providing care to patients during the COVID-19 pandemic. Quantitative data were collected using an anonymous online survey that assessed the well-being and satisfaction of licensed respiratory therapists in the state of Mississippi. More specially, survey responses ($n$ = 326) were quantitatively evaluated to measure the association between demographic variables and compassion satisfaction (CS), burnout (BO), and secondary traumatic stress (STS). Ninety-seven percent of participants reported a medium to high CS level, while 74% indicated having a medium to high level of BO, and 69% reported a medium to high level of STS. Neither age nor gender had a significant difference in CS ($p$ = 0.504; $p$ = 0.405), BO ($p$ = 0.161; $p$ = 0.285), or STS ($p$ = 0.145; $p$ = 0.252). Those working for more than 10 years at their current employer had higher CS (M = 38.7) and lower BO (M = 24.9) and STS (M = 24.8) scores. The number of hours worked, specifically overtime, had a significant impact on BO ($\beta$ = 0.09, $p$ = 0.028) and STS ($\beta$ = 0.0.11, $p$ = 0.019), but not CS ($\beta$ = 0.02, $p$ = 0.655). These findings suggest that the number of years employed in the field impacts the level of compassion satisfaction and contributes to lower levels of burnout and secondary traumatic stress. The age of a patient may also affect levels of compassion and burnout. The results of this study highlight the importance of developing incentive plans in an effort to retain employees.

**Keywords:** compassion satisfaction; respiratory therapists; burnout

## 1. Introduction

The COVID-19 pandemic had an unmeasurable impact on the United States healthcare system. During the height of the pandemic, scarce resources, limited hospital space, and staff shortages resulted in hospital policies prioritizing immediate care for the masses over achieving the best outcomes for individuals [1]. COVID-19 has strained the healthcare systems in unprecedented ways, which has led to an emphasis on telehealth and increased attention toward sustainable staffing practices [2]. In April 2023, the World Health Organization [3] reported over 100 million confirmed cases and 1,121,819 COVID-19-related deaths in the United States.

Supply chain issues, staffing shortages, and the rapid influx of acutely ill patients required providers to drastically adapt clinical care. Healthcare providers were expected to

work extra shifts and longer hours in unchartered territory. The novelty of SARS-CoV-2 further increased fear, and providers continually had to cope with devastating patient outcomes [4]. Due to the increased clinical demands during the pandemic, compassion fatigue was commonly experienced by healthcare professionals [4].

Compassion fatigue consists of two parts—burnout and secondary traumatic stress [5]. These negatively reflect the quality of life and can result in a detrimental impact on patient care and productivity [6]. In 2020, by surveying over 500 healthcare professionals, Ruiz-Fernandez et al. [7] found that compassion fatigue scores were moderate to high among both physicians and nurses. However, there is a lack of literature assessing compassion fatigue among respiratory therapists. Respiratory therapists played a critical role during COVID-19 including the management of mechanical ventilation, prone positioning, noninvasive support in direct patient care, training other healthcare professionals in the proper use of personal protective equipment (PPE), and implementing infection control protocols, to name a few [8]. Given the high respiratory acuity of many individuals infected with SARS-CoV-2, respiratory therapists were in high demand and short supply during the pandemic. These providers were continually confronted with patient loss, fear of the unknown, and risk of infection.

One recent study by Miller et al. [9] analyzed survey responses from 1114 individuals and found a burnout rate of 79% among respiratory therapists. Given the detrimental effects compassion fatigue can have on these professionals and their ability to deliver high-quality healthcare, this high rate of burnout highlights the importance of conducting further research on the well-being of this population. This study utilized the Professional Quality of Life Scale (ProQOL) developed by Stamm [5] to assess the degrees of compassion satisfaction (CS), burnout (BO), and secondary traumatic stress (STS) among certified respiratory therapists (CRTs) and registered respiratory therapists (RRTs) in the state of Mississippi.

## 2. Materials and Methods

This study received ethical approval (IRB-2022-234) from the Institutional Review Board (IRB) located at the University of Mississippi Medical Center on 18 July 2022.

### 2.1. Study Design

This research used a cross-sectional survey (Appendix A) to investigate the experiences of licensed respiratory therapists in the state of Mississippi who provided care to patients during the COVID-19 pandemic. The survey utilized the Professional Quality of Life (ProQOL) survey to assess the participants' quality of life in terms of CS, BO, and STS. Variables included demographic questions (age, gender, and education level), years of experience as a respiratory therapist, years of employment with the current employer, whether working overtime, caring for pediatric or adult patients, past COVID-19 diagnosis themselves, and Items 1–30 on the ProQOL survey.

A major strength of utilizing the ProQOL lies in the established validity and reliability of its survey items that have been widely acknowledged within the field [5]. The three steps for calculating the scores in the ProQOL manual were strictly followed. Items 1, 4, 15, 17, and 29 were reversed. The totals for each subscale (CS, BO, and STS) were summed, and z scores were converted to t scores with raw score means. The cut scores were set around the 25th and 75th percentiles, as suggested.

### 2.2. Participants

The target audience for this study was licensed CRTs and RRTs in the state of Mississippi. Permission and an email distribution list were obtained from the Mississippi State Board of Respiratory Therapy. The projected participant total was $n = 2655$, representing all licensed respiratory therapists in the state of Mississippi. Informed consent was waived by the IRB, and the return of the survey was considered the participants' consent to participate.

*2.3. Data Collection and Analysis*

Data collection was performed from September to November 2022 using an anonymous survey developed from a secure RedCap online platform, incorporating components of the ProQOL survey. The survey was distributed to the participants via email by the study personnel. The email included a link and a QR code, which the participants could use to access and complete the survey online. The survey collected data on various aspects of the participants' experiences as licensed respiratory therapists.

The collected data were analyzed with Stata 18 by our study personnel using the guidelines provided in the ProQOL Manual. Descriptive statistics, such as means, standard deviations, and frequencies, were calculated to summarize the survey responses. Inferential statistical techniques, such as Kruskal–Wallis tests, were used to examine any differences in CS, BO, and STS according to demographic and work-related characteristics. To determine the factors that contribute to CS, BO, and STS, multiple regression analyses were conducted.

The data collection process ensured complete anonymity to protect the participants' privacy. No identifying information was collected during the survey, and all responses were treated with strict confidentiality. The survey responses were stored securely and accessible only to our study personnel.

## 3. Results

The population size of licensed respiratory therapists (CRTs and RRTs) in the state of Mississippi was 2655. Based on power calculations, the optimal sample size was 336 with a 95% confidence level and a 5% margin of error. A total of 345 (13%) responded. After eliminating incomplete responses, 326 responses were included in the final analysis.

Table 1 shows the participant demographic characteristics. The ages of respiratory therapists were relatively evenly distributed with 51% being 44 years old or below and 49% being 45 or above. The majority of the participants were females (78%) and held RRT licensures (78%). The mean years of experience was 18.71 (SD = 12.03), and the majority of them had between 10 and 19 years of experience in the field (29%). The mean years with the current employer was 13.44 (SD = 10.81), with 55% having worked for their current employers for 10 years or more. Regarding whether the participants had worked overtime to care for COVID-19 patients, 85% responded "yes". The mean work hours per week was 49.26 h (SD = 15.14). The majority worked between 40 and 59 h (56%), and 30% worked 60 h or more per week. All participants (100%) indicated that they provided care for COVID-19 patients. Almost two-thirds of them (65%) provided care for adult patients, 32% for both adult and pediatric patients, and 3% for pediatric patients only. Also, over 62% of the participants were diagnosed with COVID-19 during the pandemic.

**Table 1.** Participant characteristics.

| Variables | Overall (*n* = 326) |
|---|---|
| Age category, % | |
| <35 | 67 (21%) |
| 35–44 | 96 (30%) |
| 45–54 | 77 (24%) |
| ≥55 | 78 (25%) |
| Gender, % | |
| Male | 72 (22%) |
| Female | 254 (78%) |
| Education, % | |
| CRT | 71 (22%) |
| RRT | 254 (78%) |

**Table 1.** *Cont.*

| Variables | Overall (*n* = 326) |
|---|---|
| Years of experience, mean (SD) | 18.71 (12.03) |
| Categories, % | |
| <10 | 78 (28%) |
| 10–19 | 82 (29%) |
| 20–29 | 61 (22%) |
| ≥30 | 58 (21%) |
| Years with current employer, mean (SD) | 13.44 (10.81) |
| Categories, % | |
| <10 | 87 (45%) |
| ≥10 | 108 (55%) |
| Work overtime, % | |
| Yes | 278 (85%) |
| No | 48 (15% |
| Work hours, mean (SD) | 49.26 (15.14) |
| Categories, % | |
| <40 | 32 (15%) |
| 40–59 | 121 (56%) |
| ≥60 | 65 (30%) |
| Provided care for COVID-19 patients, % | 326 (100%) |
| Care Type, % | |
| Adults | 211 (65%) |
| Pediatrics | 11 (3%) |
| Both | 104 (32%) |
| Diagnosed with COVID-19, % | |
| Yes | 203 (62%) |
| No | 123 (38%) |

Table 2 summarizes the participants' degree of compassion satisfaction, burnout, and secondary traumatic stress. The mean scores for CS, BO, and STS were 37.53 (SD = 6.98), 26.43 (SD = 6.54), and 26.57 (SD = 7.54). Scores were further categorized into low (<23), medium (23–41), and high (>42). Though 97% of participants reported medium to high CS, 74% reported having a medium to high level of BO, and 69% reported a medium to high level of STS.

**Table 2.** Degree of compassion satisfaction, burnout, and secondary traumatic stress (*n* = 326).

| Variables | Percentages | Mean (SD) | Min, Max |
|---|---|---|---|
| CS | | 37.51 (6.98) | (10,50) |
| low (<23) | 7 (2%) | | |
| medium (23–41) | 223 (68%) | | |
| high (>42) | 96 (29%) | | |
| BO | | 26.43 (6.54) | (10,44) |
| low (<23) | 86 (26%) | | |
| medium (23–41) | 235 (72%) | | |
| high (>42) | 5 (2%) | | |
| STS | | 26.57 (7.54) | (11,49) |
| low (<23) | 103 (32%) | | |
| medium (23–41) | 214 (66%) | | |
| high (>42) | 9 (3%) | | |

Note: CS = compassion satisfaction; BO = burnout; SS = secondary traumatic stress; M = mean; SD = standard deviation.

Table 3 presents Kruskal–Wallis test results showing the mean scores of CS, BO, and STS in subgroups of demographic characteristics. Our analysis did not reveal any statistically significant differences in CS, BO, and STS scores among subgroups defined by

age, gender, education, and years of experience. However, participants with less than 10 years with their current employer and those with 10 or more years showed statistically significant differences in their CS, BO, and STS scores. Those who have been with their current employers for 10 or more years showed a higher mean CS score (M = 38.7, SD = 6.2), a lower mean BO score (M = 24.9, SD = 6.3), and a lower STS mean score (M = 24.8, SD = 6.9). Whether the participants worked overtime was also found to be statistically associated with the CS, BO, and STS scores. Those who worked overtime demonstrated statistically significantly lower CS (M = 37.1, SD = 7.0) scores and higher BO (M = 27.0, SD = 6.5) and STS (M = 27.1, SD = 7.6) scores. Hours of work did not impact CS scores; however, the longer hours the participants worked, the higher the BO (<40: M = 23.5, SD = 6.4, 40–59: M = 27.1, SD = 6.4, ≥60: M = 28.2, SD = 6.6, respectively) and STS scores (<40: M = 22.9, SD = 6.2, 40–59: M = 26.9, SD = 7.3, ≥60: M = 29.2, SD = 8.4) were. The patient type did not affect the participants' CS scores, but it led to statistically significantly different BO and STS scores. Those who cared for pediatric patients showed the lowest BO (M = 23.6, SD = 5.1) and STS (M = 22.4, SD = 4.5) scores, and those who cared for both adult and pediatric patients showed the highest BO (M = 27.5, SD = 6.9) and STS (M = 27.9, SD = 8.3) scores. Lastly, whether the participants were diagnosed with COVID-19 themselves was not associated with CS, BO, and STS.

**Table 3.** Mean scores of CS, BO, and STS according to the participant demographic characteristics (*n* = 326).

| Variables | CS | | BO | | STS | |
|---|---|---|---|---|---|---|
| | Mean (SD) | *p*-Value | Mean (SD) | *p*-Value | Mean (SD) | *p*-Value |
| Age category | | 0.504 | | 0.161 | | 0.145 |
| <35 | 36.7 (7.6) | | 28.2 (7.2) | | 28.2 (8.9) | |
| 35-44 | 37.3 (6.1) | | 26.5 (6.5) | | 26.6 (7.2) | |
| 45–54 | 38.4 (7.2) | | 25.3 (5.9) | | 25.0 (7.3) | |
| ≥55 | 37.6 (7.3) | | 26.0 (6.4) | | 26.8 (6.7) | |
| Gender | | 0.405 | | 0.285 | | 0.252 |
| Male | 36.9 (7.5) | | 27.3 (6.6) | | 25.7 (7.4) | |
| Female | 37.7 (6.8) | | 26.2 (6.5) | | 26.8 (7.6) | |
| Education | | 0.139 | | 0.444 | | 0.894 |
| CRT | 36.4 (7.5) | | 26.9 (6.7) | | 26.2 (5.9) | |
| RRT | 37.8 (6.8) | | 26.3 (6.5) | | 26.7 (8.0) | |
| Years of experience | | 0.330 | | 0.123 | | 0.444 |
| <10 | 37.5 (7.0) | | 27.6 (6.3) | | 27.4 (7.6) | |
| 10–19 | 36.9 (7.4) | | 26.7 (7.6) | | 26.2 (8.2) | |
| 20–29 | 37.3 (7.7) | | 26.3 (6.2) | | 25.4 (7.4) | |
| ≥30 | 39.1 (6.2) | | 24.8 (5.7) | | 25.8 (6.7) | |
| Years with current employer | | 0.039 * | | 0.007 | | 0.003 ** |
| <10 | 36.7 (6.9) | | 27.6 (6.8) | | 28.2 (8.0) | |
| ≥10 | 38.7 (6.2) | | 24.9 (6.3) | | 24.8 (6.9) | |
| Work overtime | | 0.009 ** | | <0.001 *** | | 0.003 ** |
| Yes | 37.1 (7.0) | | 27.0 (6.5) | | 27.1 (7.6) | |
| No | 40.0 (6.2) | | 23.1 (5.9) | | 23.5 (6.2) | |
| Work hours | | 0.929 | | 0.005 | | 0.002 ** |
| <40 | 37.6 (7.2) | | 23.5 (6.4) | | 22.9 (6.2) | |
| 40-59 | 37.1 (7.1) | | 27.1 (6.4) | | 26.9 (7.3) | |
| ≥60 | 37.0 (6.9) | | 28.2 (6.6) | | 29.2 (8.4) | |
| Provided care for COVID-19 patients | 37.5 (7.0) | NA | 26.4 (6.5) | NA | 26.6 (7.5) | NA |
| Care Type | | 0.259 | | 0.039 | | 0.022 * |
| Adults | 37.4 (6.9) | | 26.1 (6.4) | | 26.1 (7.1) | |
| Pediatrics | 40.9 (5.4) | | 23.6 (5.1) | | 22.4 (4.5) | |
| Both | 37.4 (7.2) | | 27.5 (6.9) | | 27.9 (8.3) | |

**Table 3.** *Cont.*

| Variables | CS | | BO | | STS | |
|---|---|---|---|---|---|---|
| | Mean (SD) | *p*-Value | Mean (SD) | *p*-Value | Mean (SD) | *p*-Value |
| Diagnosed with COVID-19 | | 0.625 | | 0.775 | | 0.859 |
| Yes | 37.7 (6.7) | | 26.6 (7.2) | | 26.6 (7.3) | |
| No | 37.2 (7.4) | | 26.3 (6.2) | | 26.6 (7.7) | |

\* $p \leq 0.05$, \*\* $p \leq 0.01$, and \*\*\* $p \leq 0.001$.

Table 4 shows the results of the multivariate linear regression. The number of years with the current employer was found to have a statistically significant negative association with BO ($\beta = -0.13$ (95% CI, $-0.24$, $-0.01$), $p = 0.029$) and STS ($\beta = -0.14$ (95% CI, $-0.27$, $-0.01$), $p = 0.034$) but not CS ($\beta = 0.09$ (95% CI, $-0.02$, $0.21$), $p = 0.114$). Work hours was found to be positively associated with BO (($\beta = 0.09$ (95% CI, $0.01$, $0.17$), $p = 0.028$) and STS (($\beta = 0.11$ (95% CI, $0.02$, $0.21$, $p = 0.019$) but did not impact CS ($\beta = 0.02$ (95% CI, $-0.06$, $0.10$, $p = 0.655$). All other factors, such as age, gender, and education, did not exhibit a statistically significant association with CS, BO, or STS.

**Table 4.** Beta estimates from multivariate regression (beta estimate (95% CI), *p*-value).

| Variables | Beta Estimate (95% CI), *p*-Value | | |
|---|---|---|---|
| | CS | BO | STS |
| Age | | | |
| <45 | Reference | Reference | Reference |
| ≥45 | 1.23 ($-1.30$, 3.75), $p = 0.339$ | $-1.39$ ($-3.86$, 1.08), $p = 0.267$ | 0.23 $-2.66$, 3.12), $p = 0.875$ |
| Male | 0.55 ($-1.91$, 3.01), $p = 0.659$ | 0.67 ($-1.74$, 3.07), $p = 0.584$ | $-1.96$ ($-4.77$, 0.85), $p = 0.171$ |
| Education | | | |
| CRT | Reference | Reference | Reference |
| RRT | 0.34 ($-2.09$, 2.78), $p = 0.780$ | 0.61 ($-1.77$, 2.98), $p = 0.614$ | 0.96 ($-1.82$, 3.74), $p = 0.495$ |
| Years with current employer | 0.09 ($-0.02$, 0.21), $p = 0.114$ | $-0.13$ ($-0.24$, $-0.01$), $p = 0.029$ \* | $-0.14$ ($-0.27$, $-0.01$), $p = 0.034$ \* |
| Work hours | 0.02($-0.06$, 0.10), $p = 0.655$ | 0.09 (0.01, 0.17), $p = 0.028$ \* | 0.11 (0.02, 0.21), $p = 0.019$ \* |

Note. \* $p \leq 0.05$.

## 4. Discussion

This study aimed to evaluate the degree of compassion satisfaction and fatigue amongst respiratory therapists in the state of Mississippi as a result of providing care to patients during the COVID-19 pandemic. Specifically, this quantitative study attempted to measure the degree of compassion satisfaction (CS), burnout (BO), and secondary traumatic stress (STS) among study participants using the Professional Quality of Life Scale (ProQOL) developed by Stamm [5]. Prior studies have shown a significant negative impact of COVID-19 on healthcare workers' well-being ranging from compassion fatigue to low productivity [4,6].

Recent research has found that healthcare workers have experienced heightened levels of burnout and higher levels of compassion when compared with the pre-pandemic [4,10]. Our study did not find any statistically significant associations between CS, BO, or STS and gender and age. Our study did, however, find that 97% of participants reported medium to high levels of compassion satisfaction. Similar results were reported by Bahari et al. [11] and Spirczak et al. [12] who found that nurses and respiratory therapists, respectively, reported above-average to high levels of compassion satisfaction. Spirczak et al. [12] also found that younger and less experienced respiratory therapists tend to have higher levels of BO and lower CS. Additionally, we found a significant difference in CS, BO, and STS scores relevant to the number of years worked with the current employer. Participants who indicated having worked ten or more years at their current place of employment indicated higher mean CS scores, lower mean BO scores, and lower mean STS scores. This phenomenon

may be explained by the typical inclination of individuals who have stayed with the same organization to already experience a higher level of job satisfaction and lower BO and STS, as suggested by de Vries et al. [13]. These findings differ from Algamdi [14], who found that nurses with less than six months of experience with the same employer reported higher levels of compassion satisfaction. In relation to the documented elevated levels of burnout experienced due to the pandemic, a higher level of CS could also indicate that these participants have a greater support system within their organization. Profit et al. [15] previously found that healthcare providers' perceived level of burnout can be directly related to the organizational culture of their workplace. More recently, Dwyer et al. [16] found that in the midst of the pandemic, healthcare providers reported a higher sense of value in their job roles, which may have also added to the higher level of compassion satisfaction reported by the participants in this study.

In determining the association between the variables (CS, BO, and STS) and the number of hours worked per week, this study found that working overtime was statistically significantly associated with self-reported scores. Participants who indicated having worked overtime expressed lower CS with higher BO and STS. It has been reported that overtime, particularly in cases where providers feel pressured to work overtime, has a negative impact on the well-being of healthcare providers [17]. It was also found that fatigue associated with long hours of work produced an adverse effect on workers' performance and productivity [17]. This could also indicate an impact on the level of compassion demonstrated toward patients.

Although our study found no direct correlation between the types of patients being treated and compassion satisfaction, there were statistically significant differences in BO and STS scores for those who only treated pediatric patients compared with those who treated both pediatrics and adults. Participants who indicated treating only pediatric patients showed the lowest BO and STS scores, while those who indicated caring for adult and pediatric patients reported the highest BO and STS scores. In contrast, Panagou et al. [10] found moderate levels of BO and STS among providers treating pediatric Intensive Care Unit (ICU) patients, which was attributed to the severity of the patients being treated and the negative outcomes associated with pediatric patients in the ICU.

Overall, our study found that the years of employment in the same facility and the years of working in the field of respiratory therapy are statistically associated with BO and STS. It may be beneficial for organizations to establish or strengthen existing support services for respiratory therapists and other front-line healthcare workers to increase retention and quality of life while mitigating burnout and secondary traumatic stress [10].

This study has several limitations. Firstly, it was conducted as a cross-sectional survey, inherently involving a single time point of data collection, which may raise concerns about the presence of common method variance (CMV). Secondly, the data gathered relied on self-reported responses, introducing the possibility of subjectivity and potential bias in the results. Lastly, our research was specifically centered on respiratory therapists in Mississippi, thereby limiting the generalizability of the findings to a broader spectrum of healthcare providers who might have experienced compassion fatigue and reduced compassion satisfaction as a consequence of the COVID-19 pandemic. Future research should consider using longitudinal or experimental designs to better establish the relationships among variables and mitigate the potential for CMV. To gain a deeper understanding of participants' self-reported levels of CS, BO, and STS, it could also be beneficial to conduct qualitative studies in the future with not only the respiratory therapists in the state of Mississippi but other healthcare providers as well to gain a more comprehensive understanding of their well-being.

## 5. Conclusions

The COVID-19 pandemic had a significant impact on the well-being of healthcare providers. This study attempted to measure the degree of compassion satisfaction (CS), burnout (BO), and secondary traumatic stress (STS) among licensed respiratory therapists

(CRTs and RRTs) in the state of Mississippi. The findings in this study suggest that the number of years respiratory therapists are employed in the field and with their employer plays a critical role in their ability to maintain compassion satisfaction toward patients and decrease the levels of burnout and secondary traumatic stress. Previous research has found similar results in respiratory therapists and other healthcare providers, specifically nurses, in relation to a higher number of years worked in the field and higher levels of compassion satisfaction. Furthermore, the age of the patient being treated may influence the levels of compassion and burnout experienced by healthcare providers. Our findings highlight the importance of developing retention plans within healthcare institutions as an urgent measure to incentivize providers to remain employed.

**Author Contributions:** Conceptualization, D.D. and X.Z.G.; methodology, D.D.; software, X.Z.G. and X.Z.; validation, D.D., B.R., R.W., J.D., X.Z. and X.Z.G.; formal analysis, X.Z.G. and D.D.; investigation, X.Z.G., D.D. and B.R.; resources, X.Z.G.; data curation, X.Z.G. and X.Z.; writing—original draft preparation, D.D., B.R., J.D., R.W. and X.Z.G.; writing—review and editing, D.D., B.R., X.Z.G., R.W. and J.D.; supervision, X.Z.G. and D.D.; project administration, D.D. All authors have read and agreed to the published version of the manuscript.

**Funding:** This research received no external funding.

**Institutional Review Board Statement:** This study was approved by the Institutional Review Board (or Ethics Committee) of the University of Mississippi Medical Center (protocol code IRB-2022-234 and 18 July 2022).

**Informed Consent Statement:** Informed consent was waived due to the fact that requesting consent could compromise the anonymity and confidentiality of respondents.

**Data Availability Statement:** The data used and analyzed during the current study are available from the corresponding author upon reasonable request.

**Conflicts of Interest:** The authors declare no conflict of interest.

### Appendix A

Survey Link: Compassion Fatigue and Satisfaction Among Respiratory Therapists Study (umc.edu, accessed on 9 January 2022).

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
