# Peer review of "Compassion Satisfaction, Burnout, and Secondary Traumatic Stress among Respiratory Therapists in Mississippi: A Cross-Sectional Study"

_2673-527X, doi:10.3390/jor3040018_

Round 1

Reviewer 1 Report

Comments and Suggestions for Authors

I am glad to have this opportunity to review this article. After reading it, I appreciate great efforts made by the author(s). However, there are some problems. Hence, this article must have a major revision. The problems are listed as follow for your reference.

1. The data source of survey comes from the single time, which may cause CMV, but the article does not address it.

2. There is a weak reasoning in the relationships between the variables.

3. I don’t find the validity and reliability analysis of variables in this paper.

Author Response

Thank you for reviewing our manuscript. We have attached a Word document addressing Reviewer 1's comments. Please let us know if you have any questions. 

Reviewer 2 Report

Comments and Suggestions for Authors

Dear authors, congratulations on the subject of study of the manuscript, as it is of importance to study of the consecuencias sobre effect on the well-being of healthcare professionals after The COVID-19 pandemic

After reviewing the manuscript, I submit the following comments.

Best regards,

In “2.1 Study Design” subsection

In lines 81 and 82, you wrote “Professional Quality of Life (ProQOL) questionnaire to assess the participants' well-being and job satisfaction” without describing the dimensions to be studied within the questionnaire, although you name them in different sections. As you describe the variables to be studied, you must include here, both the dimensions studied in the questionnaire, as well as the cut-off scores for each dimension (which are also presented in other sections, but not in this subsection), this subsection being appropriate to describe the variables to study.

In “2.2 Participants” subsection

The period of time in which the survey data was collected is not mentioned.

In “2.3 Data Collection and analysis” section

The use of an informed consent model that participants must accept online before beginning the study questionnaire is not mentioned.

It is recommended that they mention it, if they have used it, even if the participation is online and without identification data, as they mention.

In “3. Results” section

When they present the SD data in the body of the manuscript, they must accompany it with a 95% confidence interval (95%CI), as if they specified it in other statistical calculations, throughout the entire section.

In Table 2, replace or complement said column with the result of the 95% CI

Author Response

Thank you for reviewing our manuscript. We have attached a Word document addressing Reviewer 2's comments. Please let us know if you have any questions. 

Reviewer 3 Report

Comments and Suggestions for Authors

Major comment

Based on the critical role of respiratory therapists during COVID-19, the research conception to assess the well-being and satisfaction of licensed respiratory therapists is perceptive. However, the study seems not to have obtained the respiratory therapist-specific results.

The most important finding is that the number of years employed in the field plays a significant role; however, it might not exceed the results of previous studies, “Having worked at the same organization for a significant amount of time could simply indicate a higher level of job satisfaction overall which may explain these results (de Vries et al., 2023).” In the conclusion, the authors should clearly state whether or not there were findings unique to respiratory therapists that differ from those of other professions.

Minor comment

In line number 28

p=0.162 does not match the p-value of 0.161 in Table 3.

Author Response

Thank you for reviewing our manuscript.  We have attached a Word document addressing Reviewer 3's comments.  Please let us know if you have any questions. 

Round 2

Reviewer 1 Report

Comments and Suggestions for Authors

It is good.